# The New Role of SGLT2 Inhibitors in the Management of Heart Failure: Current Evidence and Future Perspective

**DOI:** 10.3390/pharmaceutics14081730

**Published:** 2022-08-18

**Authors:** Saverio Muscoli, Francesco Barillà, Rojin Tajmir, Marco Meloni, David Della Morte, Alfonso Bellia, Nicola Di Daniele, Davide Lauro, Aikaterini Andreadi

**Affiliations:** 1Division of Cardiology, Fondazione Policlinico “Tor Vergata”, 00133 Rome, Italy; 2Department of Systems Medicine, University of Rome “Tor Vergata”, 00133 Rome, Italy; 3Division of Endocrinology and Diabetology, Department of Medical Sciences, Fondazione Policlinico “Tor Vergata”, 00133 Rome, Italy; 4Division of Internal Medicine—Hypertension, Department of Medical Sciences, Fondazione Policlinico “Tor Vergata”, 00133 Rome, Italy

**Keywords:** diabetes mellitus, SGLT2 inhibitors, heart failure, cardiovascular disease

## Abstract

The sodium-glucose transporter 2 inhibitors (SGLT2i) are a relatively new class of medication used in the management of type 2 diabetes. Recent clinical trials and research have demonstrated this class’s effectiveness in treating heart failure, since they reduce the risk of cardiovascular events, hospitalization, and mortality. The mechanism by which they do so is unclear; however, SGLT2i inhibit the tubular reabsorption of glucose, lowering the interstitial volume. This mechanism leads to a reduction in blood pressure and an improvement of endothelial function. As a result, improvements in hospitalization and mortality rate have been shown. In this review, we focus on the primary outcome of the clinical trials designed to investigate the effect of SGLT2i in heart failure, regardless of patients’ diabetic status. Furthermore, we compare the various SGLT2i regarding their risk reduction to investigate their potential as a treatment option for patients with reduced ejection fraction and preserved ejection fraction.

## 1. Introduction

Cardiovascular diseases remain the leading cause of death worldwide, with heart failure (HF) being one of the significant causes of mortality in patients with type 2 diabetes mellitus (T2D) [1].

According to the Framingham study, the patients with T2D have a two- and five-fold higher risk of developing HF in men and women, respectively, compared to the healthy population [2]. As a result, HF diminishes the quality of life and increases hospitalization, making this syndrome a growing public health matter [3]. HF is present in about 20–40% of diabetic patients [4].

The patients with HF are assessed mainly by echocardiography and their symptoms, and they are classified into three groups based on their ejection fraction (EF): reduced EF (EF < 40%; HFrEF); intermediate EF (EF between 40% and 49%; HFmrEF); and preserved EF (EF > 50%; HFpEF) [5]. The people suffering from HFrEF often show dyspnea, orthopnea, paroxysmal nocturnal dyspnea, fatigue, and ankle swelling [6]. The currently approved pharmacological treatments for HFrEF are reported in Table 1.

A new class of drugs, sodium-glucose co-transporter two inhibitors (SGLT2i), has recently been added to the treatment of HF [5,7]. According to the European Medicines Agency (EMA) and the Food and Drug Administration (FDA), all new glucose-lowering agents used in patients with T2D must demonstrate cardiovascular safety to prevent cardiovascular complications [7,8].

Therefore, the CVOTs (CardioVascular Outcome Trials) are designed to analyze the safety of glucose-lowering agents concerning three-point Major Adverse Cardiovascular events (MACE), including cardiovascular death, non-fatal myocardial infarction, and non-fatal stroke. The CVOTs relating to the SGLT2 inhibitors have shown that these agents are safe in terms of three-point MACE and may also be protective against HF-related events, independent of atherosclerotic cardiovascular disease (CVD) or the presence of HF at baseline [9,10]. SGLT2i is a relatively new class of drugs used for the management of T2D, and they have shown potential benefit in HFrEF (Figure 1 and Figure 2) [11]. 

The FDA has approved the clinical use of the following SGLT2i: Empagliflozin; Dapagliflozin; Canagliflozin; and Ertugliflozin, while to date, Ertugliflozin has not yet been authorized by the EMA [4]. These drugs were initially developed from phlorizin found in the bark of the apple tree, which acts as a non-selective competitive inhibitor of the SGLT2 protein [12]. The synthetic analogs of phlorizin were developed to have a higher selectivity for SGLT2, a longer half-life (>12 h), and better oral bioavailability [12]. This class is unique compared to any other antihyperglycemic in that they have a significant blood glucose-lowering effect, independent of insulin [13].

The SGLT2 transports a glucose molecule across the glomerular membrane by using the electrochemical gradient of Na^+^ transport [13]. These transporters are responsible for the reabsorption of about 90% of glucose in the S1 segment of the proximal tubules [13]. In healthy subjects, glucosuria occurs when the blood glucose level exceeds 180 mg/dL [14].

However, in people with diabetes, SLGT2 is upregulated, leading to an increased glucose reabsorption in the proximal tubules, and thus shifting the threshold of glycosuria to ≈ 220 mg/dL [4].

The SGLT2 activity requires energy provided by the gradient generated by the Na^+^/K^+^ ATPase at the basolateral surface of the PCT epithelial cells, in contrast to the passive transport of glucose from the cytoplasm to plasma via GLUT2 [15]. In addition, the upregulation of the SGLT2 gene and lower glucose excretion in the urine leads to worsened glycemic control [16]. SGLT2i reduces glucose reabsorption in the renal tubules and increases glucose excretion in the urine, lowers blood glucose levels, and reduces blood pressure and HbA1c by 0.5–1% or by 0.6–11 mmol/mol [17]. The selective inhibitor of SGLT2 acts on the kidney’s proximal tubule by blocking the resorption of glucose and causing glucosuria [18]. Therefore, the use of SGLT2i in T2D patients leads to glucosuria, which lowers blood glucose levels and minimizes the risk of hypoglycemia [4]. Recent evidence has documented the efficacy of SGLT2i in reducing cardiovascular complications and hospitalization in patients with and without diabetes (Figure 3) [18].

Recently, the importance of personalized medical treatment in patients with T2D or HF has become apparent. The 2022 American Heart Association guidelines (AHA) recommend SGLT2i as a first-line therapy in patients with HFrEF [19]. These drugs also have a high evidence level 2a in patients with HFpEF and HF with mildly reduced ejection fraction (HFmrEF) [19].

We aim to elucidate the efficacy and safety of SGLT2i in patients with HFrEF and HFpEF and outline the main clinical trials and recent guidelines. We also aim to understand the molecular mechanism underlying these effects.

## 2. Sglt2i Clinical Trials on Cardiovascular Disease

### 2.1. Empagliflozin

Several CVOTs were designed to evaluate the use of SGLT2i in patients with high cardiovascular risk. The first SGLT2i cardiovascular trial was EMPA-REG OUTCOME [20], which studied 7020 patients with T2DM and established atherosclerotic cardiovascular disease (ASCVD) over 3.1 years. The primary composite endpoint was major adverse cardiac events (MACE), such as cardiovascular death, nonfatal stroke, and nonfatal myocardial infarction. These showed a 14% reduction compared with the placebo group (HR, 0.86; 95% CI, 0.74–0.99; *p* = 0.04 for superiority). In addition, the empagliflozin group had a 32% risk reduction in death from all-causes, a 38% risk reduction in cardiovascular causes (HR, 0.62; 95% CI, 0.49–0.77), and a 35% relative risk reduction in hospitalization for heart failure. No significant differences were found in the myocardial infarction or stroke rates between the two groups. An increased rate of genital infection was observed among the patients receiving empagliflozin [20]. Empagliflozin reduced the overall burden of cardiovascular complications and hospital admissions in patients with type 2 diabetes and atherosclerotic cardiovascular disease [21]. The EMPEROR-REDUCED, a multicenter, randomized, double-blind, placebo-controlled trial, aimed to investigate the effect of empagliflozin in patients with established HF with an HFrEF [22]. The study included 3730 patients with or without T2DM with a chronic HF for at least three months with a left ventricular ejection fraction (LVEF) ≤ 40%, treated with optimal medical therapy.

During the 16-month study period, the primary endpoint was CV death or HFH, whether or not diabetes was present. The primary endpoint occurred in 361 of 1863 patients (19.4%) in the empagliflozin group and in 462 of 1867 (24.7%) in the placebo group (HR, 0.75; 95% [CI], 0.65 to 0.86; *p* < 0.001). Secondary endpoints, including total HFH, were lower in the empagliflozin group than in the placebo (*p* < 0.001). In addition, the decline of glomerular filtration rate was slower in the empagliflozin- than in the placebo group (−0.55 vs. −2.28 mL per minute per 1.73 m^2^ of body surface area per year, *p* < 0.001). The latter effect of empagliflozin was associated with a lower risk of severe renal impairment [23,24].

Empagliflozin has also been shown to modulate the activation of the Ca++/calmodulin-dependent kinase II, which contributes to the activation of NHE-1 in the heart. This effect in preventing calcium overload may explain why the drug prevents the time-dependent decline in systolic heart function in experimental heart failure caused by pressure overload [22,25].

In the EMPEROR-Preserved, a double-blind study, 5988 patients with HF (NYHA class II-IV) and an LVEF > of 40% were randomized to receive empagliflozin or a placebo in addition to the standard therapy [26]. Empagliflozin reduced the combined risk of cardiovascular death or HFH in the patients with HFpEF, whether or not diabetes was present.

The total number of HFH was lower in the empagliflozin than in the placebo group (*p* < 0.001). However, empagliflozin more frequently reported uncomplicated genital and urinary tract infections and hypotension [27]. Kolijn et al. studied the effects of empagliflozin on human HFpEF myocardium and obese rats with ZDF. They showed that empagliflozin reduced the myocardial inflammation and oxidative stress, improved endothelial function, and thereby reversed the pathological suppression of the NO-sGC-cGMP-PKG pathway and its downstream targets, resulting in reduced pathological cardiomyocyte stiffness. This could be a potential mechanism by which the drug could improve CV outcomes in patients with HFpEF [28].

### 2.2. Dapagliflozin

Dapagliflozin is a selective inhibitor of SGLT2 that blocks the glucose reabsorption in the proximal tubule of the kidney, promotes glucosuria, and induces clinically significant changes in the glycemic parameters in T2DM patients [29].

Dapagliflozin was evaluated in the DECLARE-TIMI 58 study, which enrolled 17,160 diabetic patients (HA1c level at least 6.5% but less than 12.0%) with or without (10,186 pts) established ASCVD and followed them for 4.2 years [30]. All of the eligible patients were 40 or older and had an estimated glomerular filtration rate (eGFR) of 60 mL/min. The patients were divided into two groups receiving dapagliflozin 10 mg or placebo. The primary efficacy outcomes were MACE and a composite of cardiovascular death or HFH [30]. Compared to the EMPA-REG OUTCOME and CANVAS, the inclusion criteria for this study indicated that participants were at a lower risk for CVD [17].

Dapagliflozin did not significantly reduce the primary composite outcome, including CV death, non-fatal MI, and non-fatal stroke (HR, 0.93; 95% CI, 0.84–1.03; *p* = 0.17), and hospitalization, but did result in a lower rate of cardiovascular death or HFH.

DECLARE-TIMI 58 was the first SGLT2i study to include HFH [30]. There was no significant difference in the various primary endpoints in the dapagliflozin group compared with the placebo group; however, only HFH was significantly reduced by dapagliflozin (HR, 0.73; 95% CI, 0.61–0.88). The secondary evidence of efficacy was renal composite and death from any cause. Dapagliflozin also reduced the incidence of HFH or CV death by 17% (HR, 0.83; 95% CI, 0.75–0.95). In the baseline population, 3.9% of patients had HFrEF, 7.7% had HFpEF, and the remaining 88.4% had no history of HF. It was observed that dapagliflozin reduced the number of hospitalizations or cardiovascular deaths more in patients with HFrEF (HR 0.62, 95% CI, 0.45–0.86) compared with those with HFpEF (HR, 0.88; 95% CI, 0.76–1.02; *p*-interaction 0.046) [30,31]. However, this study reported only a 2% reduction in CV death with dapagliflozin, compared with a 38% reduction in cardiovascular death with empagliflozin [18]. In addition, dapagliflozin showed a lower rate of adverse renal events [32].

DAPA-HF was the first outcome study designed to evaluate the effect of dapagliflozin in patients with HFrEF, with or without diabetes. In this trial, the patients with HF, NYHA class II, III, or IV, low LVEF 40% or less, and elevated levels of proBNP (≥600 pg/mL), received either dapagliflozin (at a dose of 10 mg once daily) or placebo, in addition to the recommended therapy [33]. This study included 4744 participants who were followed up for 18.2 months. The primary endpoint was an exacerbation of HF or death from CV causes. This outcome occurred in 16.3% in the dapagliflozin group and 21.2% in the placebo group (*p* < 0.001), regardless of the presence or absence of diabetes. The secondary endpoint was a composite of HFH and cardiovascular death, significantly reduced with dapagliflozin compared to placebo in those with and without diabetes.

The clinical benefit of dapagliflozin was mainly attributable to a 30% reduction in hospitalizations and urgent heart failure visits and a significant decrease in cardiovascular mortality (HR, 0.82; 95% CI, 0.69–0.98). The adverse effects associated with dapagliflozin, including volume depletion, renal dysfunction, and hypoglycemia, were not significantly different from the placebo group [33].

The PRESERVED-HF trial evaluated dapagliflozin in patients with HFpEF, a condition with few therapeutic options. This study randomized 324 patients with NYHA class II to IV, with LVEF higher than 45% for 12 weeks; in addition, 56% of the subjects had T2D, and 53% had atrial fibrillation. The Kansas City Cardiomyopathy Questionnaire (KCCQ) was the primary endpoint. Dapagliflozin improved the KCCQ by 5.8 points compared to the placebo group (*p* = 0.001) [34]. 

The DELIVER Phase III trial randomized 6263 patients with mildly reduced ejection fraction (HFmrEF) and HfpEF to receive dapagliflozin 10 mg or placebo. The primary endpoints were cardiovascular death and HF events. The secondary endpoints consisted of change from baseline in total symptom score of the KCCQ at 8 months, time to occurrence of CV death, and time to the occurrence of death from any cause [35]. The findings from the phase 3 DELIVER trial showed that dapagliflozin significantly reduced the risk for CV death or worsening HF compared vs. placebo [36].

Khalaf et al. studied cardiomyopathy in diabetic rats using dapagliflozin and crocin alone, or in combination with lactobacillus [37]. They showed a synergistic effect of the triple combination that reduced oxidation, inflammation, and apoptotic activities. In addition, dapagliflozin showed an increase in the expression of Cx 43 in the cardiomyocytes of diabetic rats by modulating the Akt/mTOR pathway [38]. Cx 43 is the major connexin expressed in the ventricle and plays a crucial role in the development of HF and arrhythmias [39]. Dapagliflozin also enhanced the biochemical indices, such as malondialdehyde and glutathione, as well as proinflammatory mediators such as NF-κB and tumor necrosis factor-α [38]. This could be a possible mechanism by which dapagliflozin could modulate cardiac remodeling and prevent arrhythmias in patients with HF [40].

### 2.3. Canagliflozin

The CANVAS (Canagliflozin Cardiovascular Assessment Study) program included CANVAS and CANVAS-R (renal), evaluating the effect of canagliflozin in 10,142 diabetic patients with or without ASCVD, with a mean follow-up of 188.2 weeks [41].

The clinical effect on each primary outcome was neutral, and the use of canagliflozin only reduced heart failure hospitalization (HFH) (HR, 0.67; 95% CI, 0.52–0.87).

The reduction in cardiovascular death or HFH appeared to be greater in patients with a history of HF (hazard ratio [HR] 0.61, 95% confidence interval [CI] 0.46–0.80) than in patients without a history of HF (HR 0.87, 95% CI 0.72–1.06; *p* for interaction 0.021). Regarding the secondary outcome, death from any cause, no superiority was observed between the canagliflozin and placebo groups. It was demonstrated that there was also no significant difference between the two groups in the fatal secondary outcome, death from any cause or cardiovascular causes (HR, 0.87; 95% CI, 0.74 to 1.01 and HR, 0.87; 95% CI, 0.72 to 1.06, respectively) [32,41].

The CREDENCE study examined the effect of canagliflozin in 4401 people with T2D and chronic kidney disease with or without CVD over 2.6 years. The primary outcome was a composite of end-stage kidney disease, a doubling of the creatinine level, or death from renal or cardiovascular causes. The relative risk of the primary outcome was 30% lower in the canagliflozin group than in the placebo group (*p* = 0.00001). HFH was reported in 4.0% of patients receiving canagliflozin, compared with 6.4% in the placebo group (*p* < 0.001). In addition, CV death or HFH occurred in 8.1% of canagliflozin patients compared with 11.5% of placebo patients (*p* = 0.01) [42].

### 2.4. Ertugliflozin

VERTIS-CV is a multicenter, double-blind trial that followed up 8246 diabetic patients with established ASCVD for 3.5 years and who were randomly assigned to receive 5 mg or 15 mg of ertugliflozin or placebo [43]. In this trial, ertugliflozin did not achieve superiority in reducing major CV or secondary composite renal events. The incidence of death from CV causes or HFH did not differ significantly between the trial groups. However, HFH was reduced by ertugliflozin, and it was reported that when ertugliflozin is used alone with the standard of care medication, it can decrease the risk of a sustained 40% decline in eGFR in patients with T2DM and established ASCVD. Overall, ertugliflozin reduced the risk for first HFH (HR, 0.70 [95% CI, 0.54–0.90]; *p* = 0.006) [43]. Indeed, a subgroup analysis suggested a benefit for HFH and HFH/CV death with ertugliflozin vs. placebo among patients with a higher risk (presence of albuminuria, higher KDIGO class). The adverse events, such as urinary infections observed with ertugliflozin, were similar to the known risks of the medicines in the SGLT2 inhibitor class. In the patients with type 2 diabetes mellitus, ertugliflozin reduced the risk of first and total HFH and total HFH/CV death, further supporting the use of sodium-glucose cotransporter 2 inhibitors in the primary and secondary prevention of HFH [44].

### 2.5. Sotagliflozin

Sotagliflozin is the most recent SGLT2i studied for safety and cardiovascular risk in diabetic patients. The SCORED study randomly enrolled 10,584 individuals with type 2 diabetes and chronic kidney disease, regardless of the presence of ASCVD (at least one major if age > 18 years, at least two minor if age ≥ 55 years), to receive sotagliflozin or placebo. A total of 31% of the participants had a history of HF [45]. The trial stopped early, after 1.3 years, due to a loss of funding due to COVID-19. To maintain the statistical power, the investigators changed the primary endpoint to CV death, HF hospitalization, and urgent visits for HF for sotagliflozin vs. placebo: 11.3% vs. 14.4% (*p* = 0.0004). This achieved significance by 95 days of follow-up [45]. Sotagliflozin is also able to reduce the gastrointestinal SGLT1 delay in glucose absorption and reduce postprandial glucose; this resulted in a 26% reduction in the primary outcome (HR, 0.74; 95% CI, 0.63–0.88) [45,46].

However, sotagliflozin was neutral compared with placebo in terms of mortality from CV causes or renal endpoint (HR, 0.90; 95% CI, 0.73–1.12) [45].

The SOLOIST-WHF study was designed to evaluate the effect of sotagliflozin in diabetic patients with HFrEF who were recently hospitalized for worsening heart failure HF [47]. SOLOIST-WHF was also terminated early because of a loss of funding, and 1222 patients were followed for nine months. This trial resulted in a 33% reduction in the primary outcome, defined as the composite outcome of CV death, HFH, and urgent visits for HF (HR, 0.67; 95% CI, 0.52–0.85). Sotagliflozin was neutral for CV mortality. However, it also showed a 30% reduction in HFH and urgent visits for HF (HR, 0.64; 95% CI, 0.49–0.83) [48].

As in the SCORED study, sotagliflozin was associated with diarrhea, which was not observed in DAPA-HF or EMPEROR-REDUCED. Therefore, this side effect could be a consequence of the inhibition of intestinal SGLT1 [46]. Another adverse effect observed with this agent was severe hypoglycemia [18].

## 3. Discussion

The mechanisms of action of SGLT2i in HF are still unclear, although the drugs have been shown to have multiple metabolic, hemodynamic, and organ-specific effects (Figure 3). In addition to reabsorbing glucose, SGLT2i affects renal function by enhancing diuresis processes, such as glycosuria, natriuresis, and uricosuria. They can lower intraglomerular pressure, which promotes the preservation of renal function [48,49]. SGLT2i may also produce pleiotropic effects, increase insulin sensitivity and glucose uptake in muscle cells, stimulate weight loss through renal caloric loss in glycosuria, and positively affect body fat distribution (Figure 3). A particular effect in patients treated with SGLT2i is an increase in plasma ketone bodies, presumably due to increased ketogenesis [50].

The increased ketogenesis could play a role in the organ-protective effect of these drugs [51]. The increased ketone body levels could be due to a systemic increase in free fatty acid (FFA) mobilization triggered by the reduction in plasma glucose and insulin caused by treatment. Although increased FFA mobilization is likely, there are currently no data in the literature that directly assess this mechanism [52]. In addition, SGLT2i appears to produce a more significant reduction in interstitial fluid. They resulted in a twofold increase in diuresis without evidence of plasma volume contraction or impaired renal function, which may prevent plasma volume depletion and subsequent hypoperfusion, as occasionally observed with the use of diuretics [53]. However, the complete mechanisms of cardioprotection induced by SGLT2i have to be clarified. An interesting hypothesis proposed in the animal models is the direct inhibition of the Na^+^/H^+^ exchanger (NHE) by SGLT2i, reducing the cardiac cytosolic Na+ and cytosolic Ca^2+^. NHE is overexpressed in both T2DM and HF. In addition, SGLT2i induce vasodilation and reduce myocardial oxidative stress in the healthy heart [54]. In the HF patients suffering from insulin resistance, FFA is used as an 80% energy source over glucose, resulting in the reduced efficiency of cardiac metabolism, suggesting that people with diabetes lack metabolic flexibility. By promoting a shift in metabolism from FFA to glucose oxidation, SGLT2i led to increased cardiac ATP production and prevented a decline in cardiac function [55]. In addition, the EMPA-HEART CardioLink-6 study and DAPA-HF showed that SGLT2i in T2D and ASCVD reduced left ventricular remodeling and improved diastolic function without negative inotropic effects [56].

SGLT2i reduced CV risk in relation to kidney function but not previous CVD status, and lower renal function was associated with a more significant reduction in HFH.

Although CREDENCE was not planned as a CVOT and therefore only 50.4% of the participants had prior CVD (compared with 40.6%, 65.6%, and 99.2% in DECLARE-TIMI 58, CANVAS, and EMPA- REG OUTCOME, respectively), MACE was twofold higher in this study than in DECLARE-TIMI 58. Therefore, the population of DECLARE-TIMI 58 had higher eGFR compared with CANVAS and EMPAREG OUTCOME.

In EMPA-REG OUTCOME, the baseline population had a higher rate of events with prior confirmed CVD than in CANVAS and DECLARE-TIMI 58. There is a need for large-scale trials of SGLT2i with appropriate inclusion and exclusion criteria and appropriate endpoints to ensure a straightforward comparison of the drugs. Among the SGLT2i trials (Table 2), CREDENCE had the highest CV event rates and DECLARE-TIMI the lowest. Despite these differences, the relative risk reductions (RRRs) for similar composite renal endpoints were externally consistent between the four studies [17]. In addition, the DECLARE-TIMI 58 study showed that the benefits of CV death or HFH were more pronounced in patients with HFrEF than in HFpEF. The benefit was particularly notable in the patients with LVEF ≤ 30% [31].

DAPA-HF and EMPEROR-REDUCED, the two trials specifically in patients with HFrEF, showed that the use of SGLT2i is associated with reduced hospitalization, regardless of patients’ diabetes status. In the DAPA-HF study, dapagliflozin was as effective in 55% of the patients without type 2 diabetes as in those with diabetes. This evidence of a cardiovascular benefit of an SGLT2i in patients without diabetes supports previous evidence that such a treatment has other beneficial effects, besides lowering blood glucose levels [42,57,58]. Similar results were found in the EMPEROR-REDUCED trial, in which empagliflozin was associated with fewer HFH and a slower decline in estimated GFR. In addition, empagliflozin was associated with a lower combined risk of cardiovascular death or HFH than placebo and a slower progressive decline in renal function in the patients with chronic HFrEF, whether or not diabetes was present [23].

The SOLOIST-WHF was the most recent outcome study to evaluate the effect of SGLT2i and sotagliflozin in HFrEF patients with diabetes hospitalized for worsening HF. The study showed that in the patients with diabetes who had worsening heart failure, the total number of cardiovascular deaths, hospitalizations, and urgent visits for HF was significantly lower with the SGLT2 and SGLT1 inhibitor sotagliflozin than with placebo.

Furthermore, in DAPA-HF, the cardiovascular deaths were lower in the dapagliflozin group than in the placebo group. The baseline characteristics of the SGLT2i trials in patients with HFrEF showed that the populations of DAPA-HF had higher eGFR, lower NT-proBNP levels, and a reduced use of ARNI and cardiac resynchronization therapy than those of EMPEROR-REDUCED and SOLOIST-WHF. The differences in these baseline risk factors may be related to the lower event rates, particularly HFH events in the study DAPA-HF.

The recent studies have demonstrated the efficacy of empagliflozin and dapagliflozin at a dose of 10 mg in improving HFH compared with standard treatment, with no significant differences between them [59].

For the first time, a recent study has documented the effectiveness of an SGLT2i in HF-hospitalized patients. The EMPULSE trial compared empagliflozin 10 mg with placebo in patients with a primary diagnosis of decompensated chronic HF regardless of LVEF. The patients were randomized in the hospital and treated for up to three months. The patients treated with empagliflozin had a higher clinical benefit than placebo (*p* = 0.0054), meeting the primary endpoint. In addition, clinical benefit was observed in both the acute de novo and decompensated chronic HF, independent of EF or the presence or absence of diabetes.

The EMPULSE study showed that empagliflozin reduced the adverse events in the acutely decompensated HF patients [60].

## 4. Conclusions

The use of SGLT2i in patients with T2DM has improved CVOTs and controlled metabolic effects. In addition, a reduction in HFH in the patients treated with SGT2i has been demonstrated in all of the CVOTs. This reduction in HFH risk was observed in both the patients with and without HF history. However, the patients with HF history accounted for only a small proportion of the COVT populations, particularly without documentation of LVEF for natriuretic peptide levels. These effects of SGLT2i on CV outcomes may not be directly related to glycemic control, suggesting that these clinical benefits may also apply to non-diabetic patients, particularly when affected by HF. The main goals of treatment for HF are to improve quality of life, reduce hospitalization, and decrease mortality. For that reason, further studies are necessary to evaluate the efficacy and safety of SGLT2i as a therapeutic option for HFrEF, particularly during the acute phases.

## Figures and Tables

**Figure 1 pharmaceutics-14-01730-f001:**
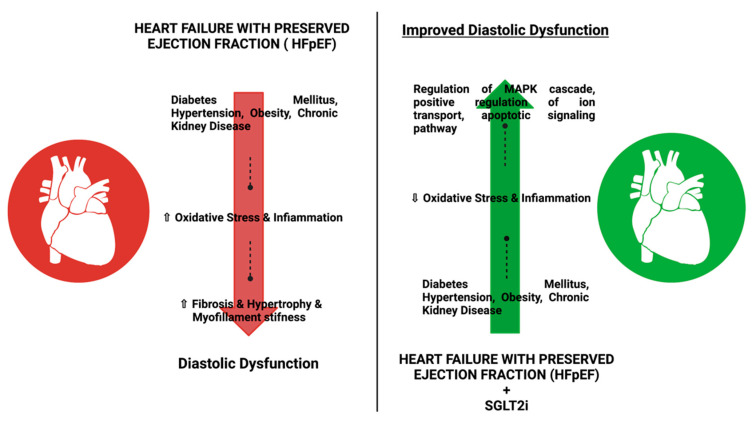
Effect of SGLT2i for the treatment of T2D in patients with HFpEF or HFrEF: The use of SGLT2i has shown improvement in diastolic dysfunction in patients with FHrEF or HF and reduced oxidative stress, infiammation, fibrosis, and myofilament stifness when compared with patients not using SGLT2i.

**Figure 2 pharmaceutics-14-01730-f002:**
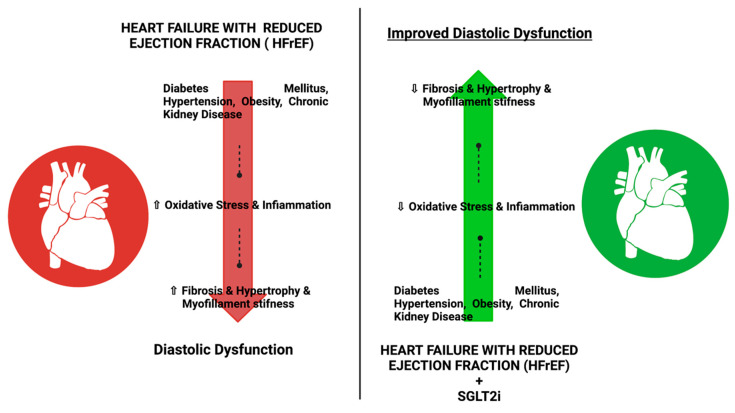
Effect of SGLT2i for the treatment of T2D in patients with HFpEF or HFrEF: The use of SGLT2i has shown improvement in diastolic dysfunction in patients with FHrEF or HF and reduced oxidative stress, infiammation, fibrosis, and myofilament stifness when compared with patients not using SGLT2i.

**Figure 3 pharmaceutics-14-01730-f003:**
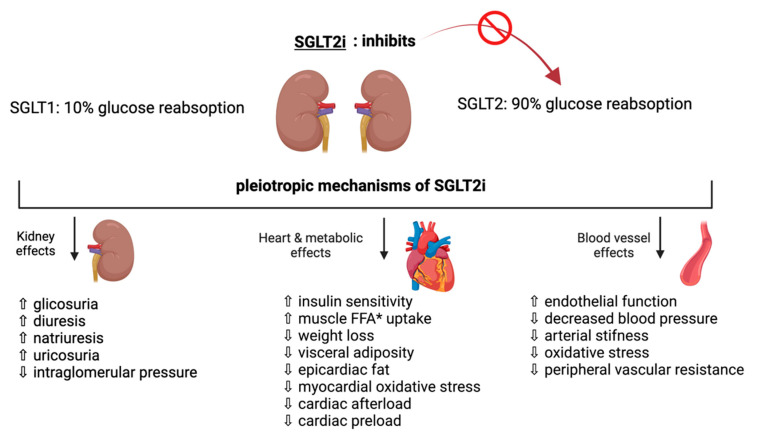
Pleiotropic effects of SGLT2i: recent evidence supports the efficacy of SGLT2i in reducing cardiovascular complication and hospitalizations in patients with and without diabetes by ameliorating renal, cardiometabolic, and vascular effects. (* FFA: free fatty acid).

**Table 1 pharmaceutics-14-01730-t001:** Class I therapy for a patient with HFrEF.

Currently Approved Pharmacological Treatment for Heart Failure Reduced Ejection Fraction (HFrEF)

(1) ACE-I * /ARBs*
(2) ARNI *: as a replacement for ACE-I
(3) ARBs: recommended for patients who cannot tolerate ACE-I or ARNI
(4) Beta-blocker
(5) MRA *
(6) SGLT2i *: Dapagliflozin / Empagliflozin
(7) Loop diuretic for fluid retention

* ACE-I: angiotensin-converting enzyme inhibitor, ARBs: angiotensin-receptor blocker, ARNI: angiotensin receptor-neprilysin inhibitor, MRA: mineralocorticoid receptor antagonists, SGLT2i: sodium-glucose transporter 2 inhibitor.

**Table 2 pharmaceutics-14-01730-t002:** Clinical Trials and effects of the SGLT2i: summary of the clinical trials of SGLT2i, most significant adverse effects, % reduction of hospitalization and primary outcome (* HFrEF: Heart Failure reduced Ejection Fraction).

SGLT2i	Trial	Patients (Number)	Duration of the Study (in Years)	Diabetes	HFrEF *	% Reduction of Primary Outcome	Adverse Effects	% Reduction in Hospitalization

Empagliflozin	EMPEROR-reduced	3730	1.4	With/without	Yes	21%	Uncompleted genital tract infection in patients treated with empagliflozin was reported more frequently compared to the placebo group. However, hypoglycemia, lower limb amputation, and bone fracture were not observed to be significantly different between the two groups.	15.4%
	EMPA-REG	7020	3.1	Yes	N/A	14%		35%
	Emperor-presrved	5988	2.4	With/without	No (LVEF >40%)	N/A		N/A

Dapagliflozin	Declare-TIMI	17,160	4.2	Yes	N/A	N/A	volume depletion, renal dysfunction, and hypoglycemia, were not reported significantly different from the placebo group	17%
	DAPA-HF	4744	1.7	With/without	Yes	21.1%		30%

Canagliflozin	CANVAS	10,142	3.6	Yes	N/A	N/A	with a higher risk of amputation primarily at the level of toe or metatarsal	14.4%
	CREDENCE	4401	2.6	Yes	N/A	N/A		37.5%

Ertugliflozin	VERTIS CV	8246	3.5	Yes	N/A	N/A	urinary infections, observed with ertugliflozin were similar to the known risks of the medicines in the SGLT2 inhibitor class.	N/A

Sotagliflozin	SOLOIST-WHF	1222	0.9	Yes	Yes	33%	Diarrhea (SGLT1 inhibition), diabetic ketoacidosis, genital mycotic infections, and volume depletion, severe hypoglycemia.	30%
	SCORED	10,584	1.3	Yes	N/A	N/A		33%

## Data Availability

Not applicable.

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
