# Peer review of "The New Role of SGLT2 Inhibitors in the Management of Heart Failure: Current Evidence and Future Perspective"

_pharmaceutics, 2022, doi:10.3390/pharmaceutics14081730_

Round 1

Reviewer 1 Report

Muscoli and colleagues have produced an interesting review summarising the applications of SGLT2i in the management of heart failure. I find this area very interesting and this article has pulled out some key studies. I have a few general comments, which might improve the article in places.

I would like to see references after the introduction to the trial, rather than at the end of the paragraph. Link to these studies and make it clear the original paper.

There is a general need for more references in the introduction, this needs to show more sources. This includes lines19 and 20 Page 2, where you are discussing a specific figure for glycosuria.

Please use international units for tests, this includes glucose and HbA1c, you can include the US units, but I would expect to see glucose as mM and HbA1c as mmol/mol alongside.

Overall there is a need to ensure that acronyms and protein names are expanded at first use. This includes Table A, which is lacking detail and proper descriptions in the caption.

It's unusual to use letters instead of numbers for the figures. Figure A is muddled, you have two captions that need to just be one and need to better describe what you are showing. 

Specific comments:

Figure B, Your arrow for weight loss would imply that there was a reduction in weight loss, you might mean to just have the arrow and weight. Thereby meaning it reduces weight.

Page 4 - lines 91-95, I assume this was left in from a draft and is not relevant? 

Page 7 line 241. I would disagree with the way this is phrased. The heart naturally oxidises more fatty acids than it does glucose. You might want to consider the wording here, to highlight that it's a greater shift towards FFA, more around the 80-90%. Or that T2DM forces a lack of metabolic flexibility.

Author Response

  • We thank the reviewer for the constructive comments. As requested we provided to add the references.
  • We thank the reviewer for the constructive comments, we added the references after the trial introduction as requested.
  • Thank you for your comment we provide to add references in the introduction
  • We thank you for your advice we have add also the international units in mmol/mol
  • We thank you for your comment, we have upgrade the table 1 with your suggestion
  • We thank you for your comment we have corrected the names of the tables and figures with numbers

Specific comments answers:

  • We thank you for your comment, as you mentioned are intention is to underline that SGLT2i should induced at the beginning weight loss as report in the studies and in our figure
  • We thank you for your clarification, yes it was a previous draft that we deleted
  • We thank you for your comment, according to your suggestion, we rephrased the text at the paper

Reviewer 2 Report

Dear Authors

In your work entitled "The New Role of SGLT2 Inhibitors in the Management of Heart Failure: current evidence and future perspective" you raise very important issues concerning the society of the 21st century. The continuous development of pharmacy helps to overcome barriers and allows the treatment of diseases that until now seemed incurable. The extension of knowledge regarding SGLT2 inhibitors and their use in the treatment of heart failure belong to a three group of compounds. In their work, the authors highlighted the results of many clinical trials that prove the effectiveness of SGLT2 inhibitors in reducing the risk of cardiovascular events. It is imperative that the exact mechanism of action of this compound be known. Thanks to this, it will be easier to apply correction and reduce the negative effects of its operation. I believe that the mauscript brings extensive knowledge in the field of pharmacy and can be successfully published in the Pharmaceutics journal.

Author Response

We thank for your comments, we provided to fix the English as requested with the help of Dr. Rojin Tamir, who is an English native speaker

Reviewer 3 Report

The manuscript titled "The New Role of SGLT-2 Inhibitors in the Management of Heart Failure: current evidence and future perspectives" is a description of the existing literature data illustrating the use of phlozins in the management of heart failure.

In the first part, the authors describe the results of large CVOT-like studies aimed at evaluating MACE and the effect of phlozins on HF. The authors synthesize a description of the pathophysiological mechanisms underlying which the effect of phlorizines in HF may lie.

The next chapter describes the effects of individual phlozin molecules on CV disease. Describing empagliflozin, they make a detailed description of the mechanism by which this molecule may act to inhibit the progression of heart failure, including results from animal models. In contrast, when describing dapagliflozin, there are no data on results from animal models, and no information is provided on the registration of this molecule in patients with heart failure and HFPEF. In general, the authors do not pay attention to this form of HF.

The strength of the paper is the presentation of the variation of the phlozin molecules in terms of the occurrence of side effects, but there is no explanation of what underlies these differences. In the discussion, the authors briefly refer to findings from animal models, which should be considered a great value of the paper. However, the rest of the discussion is a duplication of the information provided in the previous sections. Elements of suggestions for future perspectives should ring out in the discussion, which is included in the title of the paper. In conclusion, the work is worth publishing after modification with the indicated elements.

Author Response

Thank you for your comments and suggestions, we provide to improve the Dapagliflozin chapter, adding studies regarding Heart Failure with preserved ejection fraction and for the studies on the animal model.